# Injectable Polypeptide Hydrogel Depots Containing Dual Immune Checkpoint Inhibitors and Doxorubicin for Improved Tumor Immunotherapy and Post-Surgical Tumor Treatment

**DOI:** 10.3390/pharmaceutics15020428

**Published:** 2023-01-28

**Authors:** Zhixiong Chen, Yan Rong, Junfeng Ding, Xueliang Cheng, Xuesi Chen, Chaoliang He

**Affiliations:** CAS Key Laboratory of Polymer Ecomaterials, Changchun Institute of Applied Chemistry, Chinese Academy of Sciences, Changchun 130022, China

**Keywords:** hydrogel, combination therapy, immunotherapy, polypeptide, ICB therapy

## Abstract

In this work, we developed a strategy for local chemo-immunotherapy through simultaneous incorporation of dual immune checkpoint blockade (ICB) antibodies, anti-cytotoxic T-lymphocyte-associated protein 4 (aCTLA-4) and anti-programmed cell death protein 1 (aPD-1), and a chemotherapy drug, doxorubicin (Dox), into a thermo-gelling polypeptide hydrogel. The hydrogel encapsulating Dox or IgG model antibody showed sustained release profiles for more than 12 days in vitro, and the drug release and hydrogel degradation were accelerated in the presence of enzymes. In comparison to free drug solutions or hydrogels containing Dox or antibodies only, the Dox/aCTLA-4/aPD-1 co-loaded hydrogel achieved improved tumor suppression efficiency, strengthened antitumor immune response, and prolonged animal survival time after peritumoral injection into mice bearing B16F10 melanoma. Additionally, after injection of Dox/aCTLA-4/aPD-1 co-loaded hydrogel into the surgical site following tumor resection, a significantly enhanced inhibition on tumor reoccurrence was demonstrated. Thus, the polypeptide hydrogel-based chemo-immunotherapy strategy has potential in anti-tumor therapy and the prevention of tumor reoccurrence.

## 1. Introduction

Malignant tumor is a global threat to the health and life of humans [1]. The commonly used clinical treatment approaches include surgical resection, chemotherapy, radiotherapy, and immunotherapy [2,3]. Immunotherapy is a rapidly developing cancer treatment technology recently, including immune checkpoint blocking agents, immune adjuvants, chimeric antigen receptor T (CAR-T) cells, and cancer vaccines [3]. Immune checkpoints are key regulators for immunological tolerance, which can protect tumors from the recognition and attack of the immune system. In the past decade, immune checkpoint blockade (ICB) therapy has witnessed great progress in cancer therapy [4,5,6]. Cytotoxic T-lymphocyte-associated protein 4 (CTLA-4) and programmed cell death protein 1 (PD-1) are typical immune checkpoint transmembrane proteins that play key roles in different immunosuppressive processes. CTLA-4, a transmembrane protein expressed on T cells, is able to inhibit antigen presentation from antigen-presenting cells (APCs) to T cells and, therefore, restrain successive activation of T cells, through competitively binding with B7 molecules on APCs [7]. In the effector stage, the tumor recognition and killing process by effector T cells can be inhibited by the binding of PD-1 on T cells to ligands (PD-L1 and PD-L2) on tumor cells. Thus, the ICB therapy based on antibodies that block the CTLA-4 and/or PD-1 pathways has achieved remarkable clinical outcomes recently in the treatment of several types of cancers [8].

It is worth mentioning that the anti-cancer efficiency of ICB therapy is still limited by some major shortcomings, especially the insufficient objective response rates (ORRs) and serious immune-related adverse events (irAEs) [8]. It has been found that the cooperative inhibition of the CTLA-4 and PD-1/PD-L1 pathways with anti-CTLA-4 (aCTLA-4) and anti-PD-1 (aPD-1) antibodies lead to significantly enhanced ORRs [9,10,11,12,13,14,15,16,17,18,19]. In addition, it has been shown that the anti-tumor immune response can be improved through the combination with other therapy approaches, such as chemotherapy and radiation therapy [20,21]. For instance, several chemotherapeutics such as anthracyclines, oxaliplatin and cyclophosphamide can induce immunogenic cell death (ICD) of tumor cells, leading to the generation of tumor-associated antigens (TAAs) and damage-associated molecular patterns (DAMPs) [22]. Subsequently, the release of TAAs and DAMPs can stimulate an antigen-specific immune response against tumors.

On the other hand, to reduce systemic irAEs of ICB therapy, localized delivery systems based on injectable hydrogels have attracted considerable attention in recent years [23,24,25,26]. Injectable, biodegradable hydrogels have shown several advantages as topical delivery systems, including facile encapsulation of both small-molecule drugs and biomacromolecules, minimally invasive drug administration, good biocompatibility, as well as prolonged drug release behavior at targeted sites [27,28]. After the injection of drug-loaded hydrogels to the tumor sites, the ICB antibodies and combined drugs can be released locally and sustainedly, which may lead to persistently elevated drug concentration at the tumor sites while reducing systemic side effects [29,30]. Thus, several hydrogel systems loaded with an ICB antibody and a chemotherapy drug have been developed recently for improved antitumor chemo-immunotherapy [31,32,33,34,35,36]. Nevertheless, a study on the localized co-delivery of dual or multiple ICB antibodies and chemotherapeutics with injectable hydrogels has not been reported yet.

In this work, an injectable drug depot based on a thermosensitive polypeptide hydrogel was developed for the topical co-delivery of aCTLA-4, aPD-1 and an anthracycline chemotherapy drug, doxorubicin (Dox), for antitumor chemo-immunotherapy (Figure 1). As the hydrogel precursor, the methoxy poly(ethylene glycol)-*block*-poly(γ-ethyl-L-glutamate) (mPEG-PELG) aqueous solutions presented sol–gel phase transitions with the temperature rising from room temperature to 37 °C. This property facilitated the mixing of antitumor agents with the solutions at a low temperature, and spontaneous formation of drug-loaded hydrogels at the physiological temperature. The sustained release of Dox and a model antibody from the hydrogel was investigated in PBS with or without proteinase K. The capacity of the Dox-loaded hydrogel to cause ICD in B16F10 melanoma cells was revealed by testing the expression of calreticulin (CRT). The antitumor efficacy and systemic side effects of the aCTLA-4/aPD-1/Dox co-loaded hydrogel in vivo were evaluated by peritumoral injection of the hydrogel into C57BL/6 mice bearing B16F10 melanoma. The immune response of the treatment was investigated by analyzing immune cells and pro-inflammatory cytokines. Additionally, the inhibition of tumor reoccurrence after tumor resection surgery by the treatment of the hydrogel depot was further studied in a tumor resection model of melanoma-bearing mice.

## 2. Materials and Methods

### 2.1. Materials

Amino-terminated monomethyl poly(ethylene glycol) (mPEG-NH_2_, *M*_n_ = 2000) was bought from Jemkem Inc., China. Tetrahydrofuran (THF), N,N-dimethylformamide (DMF) and toluene were refluxed with CaH_2_ and distilled under N_2_ before use. γ-Ethyl-L-glutamate N-carboxyanhydride (ELG NCA) was prepared according to our previous method [37]. All the other chemical reagents were bought from Sinopharm Chemical Reagent Co., Ltd. (Shanghai, China), China and used as obtained.

InvivoMab anti-mouse PD-1 (CD279) and invivoMab anti-mouse CTLA-4 (CD152) were purchased from Bioxcell Inc. (West Lebanon, NH, USA). Dox was purchased from Beijing HVSF United Chemical Materials Co., Ltd. (Beijing, China). CD3-FITC, CD4-PE and CD8-APC antibodies were bought from Biolegend Inc (San Diego, CA, USA). Treg cell test kit was bought from Ebioscience Inc. (San Diego, CA, USA). ELISA kits for the detection of IL-2, TNF-α and IFN-γ were obtained from Shanghai Lengton Bioscience Co., Ltd. (Shanghai, China).

### 2.2. Characterization

^1^H NMR spectra of mPEG-PELG solutions in deuterated trifluoroacetic acid (CF_3_COOD) were recorded on a Bruker AV 500 NMR spectrometer (Bruker Optik GmbH, Ettlingen, Germany). Number-average molecular weight (*M*_n_) and polydispersity index (PDI) were examined by gel permeation chromatography (GPC), which was equipped with 2 Styragel^®^ HMW 6E columns (7.8 mm * 300 mm) and Waters 515 HPLC pump with a Waters 2414 refractive index detector (Waters, Milford, MA, USA). The eluant was DMF containing 0.05 M LiBr at a flow rate of 1.0 mL/min at 50 °C. Monodispersed poly(methyl methacrylate) standards were used to generate the calibration curve. The ellipticity of polymer aqueous solution (0.05 wt%) was obtained on a Chirascan CD spectrometer (Applied Photophysics, Leatherhead, UK) as a function of temperature in the range of 10–60 °C. The microstructure of freeze-dried hydrogel sample was observed by field emission scanning electron microscopy (SEM, Gemini 2, Carl Zeiss, Oberkochen, Germany).

### 2.3. Synthesis of mPEG-PELG

The mPEG-PELG block copolymer was synthesized via ring-opening polymerization (ROP) of γ-ethyl-L-glutamate N-carboxy anhydride (ELG NCA) using mPEG-NH_2_ as the macroinitiator (Appendix A) [37]. Briefly, mPEG-NH_2_ (2 g, 1.0 mmol) was dissolved in toluene (150 mL) and underwent azeotropic distillation to remove the trace water. Anhydrous DMF (50 mL) and ELG NCA (3.0 g, 15.0 mmol) were then added, and the mixed solution was allowed to stir under N_2_ atmosphere at room temperature for 3 days. After the reaction, the crude product was obtained by precipitation into cold diethyl ether and filtration. The product was then re-dissolved in DMF, purified by dialysis against deionized water for 3 days, and collected by lyophilization.

### 2.4. Phase Diagram

The sol–gel transition phase diagram of mPEG-PELG solutions was measured by the tube inversion method. The copolymer was dissolved using phosphate-buffered saline (PBS, 0.01 M, pH 7.4) at a given concentration in test tube with an inner diameter of 11 mm, and stirred at 0 °C for 72 h. The sol–gel status of the solution was observed with increasing the temperature at a rate of 2 °C per 10 min. The gelation temperature was determined when the solution kept no flow within 30 s after inverting the test tube.

### 2.5. Rheological Test

Rheological experiments were tested by placing 300 μL of copolymer solution between the plates with a diameter of 25 mm and a gap of 0.3 mm on the MCR 301 Rheometer (Anton Paar GmbH, Graz, Austria). A thin layer of silicone oil was used to seal the outer edge of the sample for preventing water evaporation. The data were tested with a fixed strain of 1% and a frequency of 1 Hz. The heating rate was set as 0.5 °C/min.

### 2.6. In Vitro Gel Degradation

The copolymer was dissolved in PBS at a concentration of 6 wt% and the solution was allowed to stir at 0 °C for 72 h. Then, 300 μL copolymer solution was placed into a glass vial of 11 mm in inner diameter and incubated at 37 °C for 10 min to obtain a hydrogel sample. Subsequently, 3 mL PBS with or without 5 U/mL proteinase K was added into the vials as the degradation medium and the hydrogels were incubated in an oscillating incubator (60 rpm) at 37 °C. At each preset time point, the solution was removed by gentle suction out from the vial with a pipette and the residual solution at the surface of the sample was further removed by a filter paper. The remaining mass of the hydrogel was weighed. Fresh medium was added to the vial. Three parallel samples were tested at each time point.

### 2.7. In Vitro Drug Release

The copolymer was dissolved in PBS with a concentration of 6 wt% and the solution was allowed to stir at 0 °C for 36 h. Dox or IgG was added to the copolymer solution, with a Dox or IgG concentration of 1 mg/mL. The solution was stirred for an additional 12 h. Then, 300 μL mixture was placed into a glass vial of 11 mm in inner diameter and incubated at 37 °C for 10 min to obtain a hydrogel sample. Following this, 3 mL PBS with or without proteinase K (5 U/mL) was added into each vial as the release medium, and all the vials were incubated in an oscillating incubator (60 rpm) at 37 °C. At a given time point, the release medium was collected to determine the amounts of released Dox and IgG, and an equal volume of fresh PBS with or without proteinase K was supplemented. The amount of Dox was quantified by a fluorophotometer with excitation and emission wavelengths of 480 nm and 592 nm, respectively. The procedure of the IgG content measurement experiment followed the instructions given by the IgG ELISA Kit (Invitrogen, Carlsbad, CA, USA). Three parallel samples were tested at each time point.

### 2.8. In Vivo Gel Degradation

Sprague Dawley rats were anesthetized and each rat was injected subcutaneously with 300 μL 6 wt% mPEG-PELG solution to form hydrogels in situ. The rats were sacrificed at predetermined intervals and dissected for observing the degradation grade of hydrogels.

### 2.9. ICD Assay

B16F10 cells were incubated in 6-well plates (3 × 10^5^/well) for 12 h, with 3 mL complete medium in each well. Then, 300 μL mPEG-PELG hydrogel loaded with Dox (5 μg/mL) was added and incubated for another 4 h. The same dose of Dox solution was used as the positive control. After digestion with 1 mL trypsin, the cells were harvested and washed, and incubated with 100 μL PBS plus 2 μL Alexa Fluor^®^ 488 calreticulin polyclonal antibody solution (Abcam, Cambridge, UK) for 1 h. Finally, the treated cells were detected and analyzed by flow cytometry (BD FASCCelesta^TM^ Flow Cytometer, BD Biosciences, San Jose, CA, USA).

### 2.10. In Vivo Antitumor Tests

B16F10 cells (1 × 10^6^) were injected subcutaneously into each C57BL/6 mouse to establish a B16F10-melanoma bearing mice model to evaluate the antitumor efficiency in vivo. The tumor-bearing mice were randomly divided into 7 groups one week after inoculation of melanoma cells (*n* = 5), as follows: PBS-treated group (denoted as PBS), free aCTLA-4 + aPD-1 mixed solution group (aCTLA-4 + aPD-1), aCTLA-4/aPD-1 co-loaded hydrogel group (aCTLA-4 + aPD-1)@gel, Dox solution group (Dox), Dox-loaded hydrogel group (Dox@gel), free Dox + aCTLA-4 + aPD-1 solution group (Dox + aCTLA-4 + aPD-1), and Dox/aCTLA-4/aPD-1 co-loaded hydrogel group (Dox + aCTLA-4 + aPD-1)@gel. The volume of paratumorally injected solution in each group was 50 μL. The dosages of Dox, aCTLA-4 and aPD-1 were controlled as 4 mg/kg, 1 mg/kg and 2 mg/kg, respectively, in the formulations containing each ingredient. The average melanoma volume and body weight were recorded and calculated in each group at predetermined time points. The melanoma volume was calculated according to the following equation: V (mm^3^) = L × W^2^/2, where L (mm) means the length of melanoma and W (mm) represents the width of melanoma. The mice were euthanized when the volume of tumor was higher than 1500 mm^3^, except those that died naturally. The survival time for all the treated groups was investigated in a separate study (*n* = 7).

### 2.11. Tumor Reoccurrence Inhibition after Tumor Resection

To evaluate the tumor reoccurrence inhibition efficacy after tumor resection, the B16F10 melanoma-bearing mice model was established in a similar procedure. After 12 days of melanoma incubation, tumor-bearing C57BL/6 mice with an average tumor volume of around 300–400 mm^3^ were chosen for establishing the tumor-resection model. The mice were randomly divided into three groups with 12 mice in each group, including PBS group, free Dox + aCTLA-4 + aPD-1 mixed solution group (Dox + aCTLA-4 + aPD-1) and Dox/aCTLA-4/aPD-1 co-loaded hydrogel group (Dox + aPD-1 + aCTLA-4)@gel. After being anesthetized with 100 μL pentobarbital (1%), each melanoma-bearing mouse was excised ~90% of melanoma, surgically sutured and subsequently treated with subcutaneous injection of the given formulation at the surgical site. The volume of hydrogel and the dosages of drugs were the same as before. The volumes of melanoma and body weights of the animals were recorded at predetermined time points. At day 12, five mice in each group were randomly selected for analysis of immune cells, and the rest of the mice were continuously monitored for evaluation of survival time (*n* = 7).

### 2.12. Cytokines Assay

The serum from C57BL/6 mice of each group was collected, and the immune related cytokines including IL-2, TNF-α, IFN-γ were tested according to the specifications of the corresponding assay kit.

### 2.13. Immune Cell Assay

Spleens, lymph nodes and tumors were collected after sacrificing the animals and processed into single cell suspensions for analysis of CD4^+^ cell, CD8^+^ cell, Treg. The whole procedure of the mentioned immune cell processing was carried out according to the specifications of the test kit. Analysis of the stained immune cells was performed by flow cytometry.

### 2.14. Organ Damage Assay

To evaluate the safety of the treatments, the main organs, such as the lung, kidney, heart, liver and spleen, were collected after the mice were sacrificed. These organs were fixed by 4% paraformaldehyde for H&E staining analysis.

### 2.15. Animal Procedure

All animal procedures were conducted in accordance with the Guidelines for Care and Use of Laboratory Animals of Jilin University and approved by the Animal Ethics Committee of Changchun Institute of Applied Chemistry, CAS (No. 2020-010). 

### 2.16. Statistical Analysis

All values were presented as mean ± standard deviation. Statistical analysis was performed by one-way ANOVA using the LSD or Tukey posttest for multiple comparison. Survival periods of animals were presented using a Kaplan−Meier curve, and analyzed by the log-rank test. Statistical significance is indicated as * *p* < 0.05, ** *p* < 0.01, and *** *p* < 0.001.

## 3. Results and Discussion

According to our previously reported method [37], mPEG-PELG was prepared via the ROP of ELG-NCA using mPEG-NH_2_ as the macroinitiator. The chemical structure and molecule weight of the obtained block copolymer were identified by ^1^H NMR and GPC, respectively (Appendix A). The degree of polymerization (DP) of mPEG-PELG was estimated as approximately 18 by comparing the integration of the peak at 0.82 ppm ascribed to the pendant methyl group of ELG residues with that at 3.09 ppm assigned to the terminal methyl group of mPEG in the ^1^H NMR spectrum. Additionally, the monomodal distribution of the resulting copolymer in the GPC trace confirmed the successful synthesis of the diblock copolymer. The *M*_n_ was determined as 4100 Da with a PDI of 1.2 according to GPC data. 

It was found that the mPEG-PELG aqueous solutions showed a sol–gel phase transition with increasing temperature, which was dependent on the polymer concentration. To investigate the gelation temperature range of the mPEG-PELG aqueous solutions, the thermo-induced sol–gel phase transitions of the polymer solutions were tested. As shown in Figure 1A, the mPEG-PELG solutions in PBS with polymer concentrations of 4~8 wt% exhibited sol–gel phase transitions with the increase in temperature, and the gelation temperature declined from 38 °C to 24 °C as the polymer concentration increased from 4 wt% to 8 wt% (Figure 1B and Appendix A). The thermo-induced gelation of the mPEG-PELG solutions may be owing to the cooperative effects of partial dehydration of mPEG blocks and the maintained β-sheet conformation at elevated temperatures (Appendix A) [38,39]. The partial dehydration of mPEG segments may result in enhanced chain entanglement and aggregation of the mPEG-PELG micelles in aqueous solution, and the β-sheet conformation of the polypeptide blocks facilitates the formation of intermolecular hydrogen bonding. These effects promote the formation of a physical crosslinking network and hydrogel. Considering the suitable gelation temperature (~30 °C), the 6 wt% mPEG-PELG solution in PBS was chosen for fabrication of the injectable hydrogel. Based on the rheological test, the 6 wt% mPEG-PELG solution showed an abrupt increase in the storage modulus (G′) with increasing temperature over 20 °C, indicating the hydrogel formation (Figure 1C). Moreover, the mixing of Dox and a model antibody, IgG, with the mPEG-PELG solution showed no obvious influence on the variations of G′ and G″ with the temperature increase, indicating that the Dox and antibody encapsulation did not markedly affect the thermo-induced gelation behavior of the polypeptide solution. Additionally, the freeze-dried hydrogel sample indicated a porous structure in the SEM image (Figure 1D), which may facilitate the encapsulation and transportation of the drugs and bioactive agents [27,28].

The degradation of hydrogels plays an important role in affecting the release behaviors of encapsulated agents. The degradation behavior of the mPEG-PELG hydrogels was evaluated in vitro and in vivo, respectively. As shown in Figure 2A, the hydrogel showed ~30% mass loss in 15 days in PBS, which may be mainly due to the surface erosion of the hydrogel. When proteinase K was added, the hydrogel degradation was obviously accelerated with ~60% mass loss within 15 days. This is likely due to the enzymatic cleavage of the polypeptide segments in addition to the hydrogel erosion [40,41]. Moreover, after subcutaneous injection into rats, the 6 wt% hydrogels degraded continuously in the subcutaneous layer of rats and completely disappeared within 5 weeks in vivo (Appendix A). It is known that there are different enzymes, such as cathepsin B, cathepsin C and elastase, in the subcutaneous layer of mammals [42]. Therefore, the degradation of the hydrogel in the subcutaneous layer of rats may be accelerated by enzymatic hydrolysis in vivo. These results indicated that the hydrogels underwent continuous degradation in vitro and in vivo.

Further, the release behaviors of chemotherapeutic drug and antibodies from the mPEG-PELG hydrogel were measured in vitro using Dox and IgG as the model drug and antibody, respectively. It was found that the Dox-loaded hydrogel showed a burst Dox release in the first 2 days, but exhibited more continuous and slower Dox release rates during the subsequent 13 days (Figure 2B). The two-stage release profile of Dox may be due to the rapid diffusion of Dox near the hydrogel surface at the first stage, and subsequent combined mechanisms of retarded Dox diffusion and slow hydrogel degradation [35]. In contrast, the Dox release was obviously accelerated with the addition of proteinase K, likely attributed to increased hydrogel disintegration in the presence of enzyme. Additionally, the release of IgG from the hydrogel was slower than Dox. This may be due to the slower diffusion rate of the antibody from the hydrogel than the small molecule (Figure 2C). The in vitro drug release data in the first 6 days were fitted to different kinetic models, including the zero-order model, first-order model, Korsmeyer–Peppas model and Higuchi model [43,44,45] (Appendix A). Through comparing the R-squared value, the Dox release profile from the hydrogel in PBS showed a relatively better degree of fitting to the Korsmeyer–Peppas model. It is noteworthy that the R-squared value was reduced for Dox release in the presence of proteinase K. Additionally, the IgG release from the hydrogel did not fit well with these kinetic models, which may be attributed to the fact that the IgG release behavior was influenced by complicated interactions between the protein and hydrogel network [46]. Therefore, the drug release tests in vitro suggested that the drug-loaded hydrogels exhibited continuous release profiles of both Dox and antibodies, which could be adjusted by the hydrogel degradation.

Calreticulin (CRT) is a typical marker on cells undergoing immunogenic cell death (ICD), which can promote antigen processing and presentation for adaptive immune response [47]. To reveal the ability of the drug-loaded hydrogel to induce ICD of tumor cells, the CRT expressions on the B16F10 melanoma cells after different treatments were tested (Figure 2D). It was found that incubation of tumor cells with either free Dox or Dox-loaded hydrogel caused enhanced CRT expression of B16F10 cells, indicating the occurrence of ICD in tumor cells.

To investigate the combination effects of Dox-mediated ICD of tumor cells and ICB therapy, Dox, aCTLA-4 and aPD-1 were co-loaded into the mPEG-PELG hydrogel to construct a depot for sustained antitumor chemo-immunotherapy. Different drug-containing systems were then peritumorally injected into C57BL/6 mice bearing B16F10 melanoma for evaluating the antitumor efficacy in vivo. The tumor-bearing mice were assigned into seven groups at random including PBS group, aCTLA-4 + aPD-1 group, (aCTLA-4 + aPD-1)@gel group, Dox solution group, Dox@gel group, Dox + aCTLA-4 + aPD-1 mixed solution group, and (Dox + aCTLA-4 + aPD-1)@gel group. It was observed that all the formulations containing Dox at a dose of 4 mg/kg exhibited an inhibition effect on tumor growth (Figure 3B). Moreover, the Dox/aCTLA-4/aPD-1 co-loaded hydrogel exhibited significantly improved tumor suppression efficiency compared to the formulations (hydrogels or solutions) containing either Dox or antibodies. Additionally, the Dox/aCTLA-4/aPD-1 co-loaded hydrogel led to a significantly extended animal survival time compared to single or multiple antitumor agent solutions, or hydrogels loaded with Dox or antibodies (Figure 3D). The results suggested that the sustained co-delivery of Dox, aCTLA-4 and aPD-1 using the hydrogel showed the best antitumor efficacy in vivo.

Additionally, it was found that all the local treatments resulted in no obvious effect on the body weight in the mice (Figure 3C), indicating a reduced systemic side effect. After treatment for 14 days, the main organs of the mice, such as the heart, liver, spleen, lung and kidney, were taken and stained with H&E staining. Observation and pathological analysis were carried out under a microscope. As shown in Figure 4, no obvious pathological changes were found in any of the groups. This confirmed that the treatments with localized injection of drug-containing formulations showed no obvious toxic side effects to normal organs. Compared to systemic administration, the local and sustained release of Dox and ICB antibodies at tumor sites can markedly reduce the blood drug concentration and drug distribution in normal tissues [29]. Thus, low systemic toxicity was observed for the local treatments.

To analyze the immune response of the mice following treatments, typical immune cells and pro-inflammatory cytokines were evaluated. It was observed that the ratios of CD8^+^ T cells in the spleen, lymph nodes and tumors were enhanced after the therapy with Dox/aCTLA-4/aPD-1 co-loaded hydrogel in Figure 5A–C. In addition, Dox/aCTLA-4/aPD-1 co-loaded hydrogel treatment reduced the ratio of immunosuppressive regulatory T cells (Tregs) in CD4^+^ cells in tumors (Figure 5D). This indicated that the continuous, simultaneous release of Dox, aCTLA-4 and aPD-1 promoted the generation and tumor accumulation of tumor-killing CD8^+^ T cells, and inhibited the accumulation of Tregs at tumor sites. 

It has been established that CTLA-4 acts on immunosuppressive functions in several aspects [7,48]. First, it inhibits antigen presentation from DCs to T cells through competitively binding CD80/CD86. Second, CTLA-4 is responsive for the functions of Tregs. Thus, the blockade of CTLA-4 by the sustained release of aCTLA-4 near the tumor site promoted the presentation of TAAs, which was generated by Dox-mediated ICD of tumor cells [22], resulting in enhanced T cell activation (Figure 1). Moreover, the ratio of Tregs at the tumor site was also reduced. In addition, PD-1, another immune checkpoint receptor usually expressed on activated T cells, also acts as a “brake” during the tumor recognition and killing, through binding to its ligand (PD-L1) expressed on tumor cells, resulting in the immune escape of tumor cells [49]. Therefore, the inhibition of the PD-1/PD-L1 pathway by a localized release of aPD-1 near the tumor site led to enhanced antigen recognition and tumor killing by CD8^+^ cytotoxic T lymphocytes (CTLs). 

Moreover, the pro-inflammatory cytokines, including IL-2, TNF-α and IFN-γ, in the blood were examined. It was observed that the concentrations of pro-inflammatory cytokines were significantly enhanced after the treatment with the hydrogel loading Dox, aCTLA-4 and aPD-1 (Figure 6). IL-2 is a cytokine that undertakes immunoregulatory functions on T cells, NK cells and NK-T cells [50]. IL-2 plays a role in promoting the proliferation of lymphocytes including T cells and NK cells. TNF-α is able to exert antitumor effects by direct showing cytotoxicity on tumor cells and inducing an antitumor immune response. IFN-γ is a crucial cytokine for cell-mediated immunity, which plays key roles in stimulating antigen presentation, cytokine secretion, as well as the activation of macrophages, NK cells and neutrophils. Overall, the localized co-delivery of Dox, aCTLA-4 and aPD-1 from the hydrogel depot resulted in a significantly enhanced antitumor immune response, which may contribute to the increased tumor inhibition efficacy in vivo.

To further evaluate the ability of the Dox/aCTLA-4/aPD-1 co-loaded hydrogel to inhibit post-operative tumor reoccurrence, a tumor resection model was established using melanoma-bearing C57BL/6 mice. As shown in Figure 7, after ~90% of the tumor was resected, the Dox/aCTLA-4/aPD-1 co-loaded hydrogel was injected into the surgical site, with PBS or the mixed solution of the multiple agents as the controls. It was observed that the remaining tumors regrew rapidly after the tumor resection surgery with additional PBS treatment (Figure 7B). The local injection of Dox/aCTLA-4/aPD-1 mixed solution at the surgery site resulted in the effective inhibition of tumor reoccurrence. Notably, the local treatment of the Dox/aCTLA-4/aPD-1 co-loaded hydrogel showed a significantly stronger inhibition effect on the tumor reoccurrence, compared to either PBS or Dox/aCTLA-4/aPD-1 mixed solution. Moreover, the Dox/aCTLA-4/aPD-1 co-loaded hydrogel therapy led to a significantly extended survival time compared to other treatments (Figure 7D). The higher efficiency in tumor reoccurrence inhibition of the Dox/aCTLA-4/aPD-1 co-loaded hydrogel may be attributed to the sustained and prolonged release manners of the chemotherapy drug and ICB antibodies from the hydrogel depot [29,30]. 

In addition, the immune cells in the spleens, lymph nodes and tumors of the treated mice were analyzed at day 12 post-surgery. It was observed that the Dox/aCTLA-4/aPD-1 co-loaded hydrogel treatment enhanced the ratio of CD8^+^ T cells in the spleen and lymph node, and caused a decrease in the ratio of Treg cells in the tumor (Appendix A). Thus, the results suggested that the Dox/aCTLA-4/aPD-1 co-loaded hydrogel can effectively inhibit post-operative tumor reoccurrence and extend animal survival time, through strengthening the antitumor immune response.

In recent studies, hydrogel-based local delivery systems have been investigated for the co-delivery of chemotherapeutics and single ICB antibody, aPD-1 or anti-PD-L1, as a strategy for local chemo-immunotherapy [31,32,33,34,36]. It has been found that the chemotherapy-mediated ICD of tumor cells could promote the antitumor immune response when combining with ICB blockade. Moreover, some agents for modulating the immunosuppressive tumor micro-environment, e.g., indoleamine-(2,3)-dioxygenase (IDO) inhibitors, were further incorporated into the hydrogel depots for improving the antitumor efficacy [35,51]. In this study, a new formulation of a hydrogel depot co-loaded with dual ICB antibodies, aCTLA-4 and aPD-1, and a chemotherapeutics was developed for topical antitumor chemo-immunotherapy. The sustained release of aCTLA-4 at the tumor site can strengthen the antigen presentation of DCs to T cells, and the simultaneous delivery of aPD-1 is able to enhance the subsequent tumor recognition and killing. Based on the treatment of B16F10 melanoma-bearing mice and post-operative mice models, the Dox/aCTLA-4/aPD-1 co-loaded hydrogel showed a significantly increased antitumor immune response, enhanced tumor inhibition efficacy and prolonged animal survival time. Additionally, it was demonstrated that the local co-delivery of the ICB antibodies and chemotherapeutics caused no obvious systemic side effects, compared to the risk of severe irAEs by systemic administration of ICB antibodies [5].

## 4. Conclusions

In the present study, an antitumor chemo-immunotherapy system was prepared by the incorporation of Dox, aCTLA-4 and aPD-1 into thermosensitive mPEG-PELG hydrogel. The polypeptide hydrogel exhibited enzyme-accelerated degradation behavior in vitro for over 15 days, and gradually degraded in the subcutaneous layer of mice. The sustained release of Dox or IgG model antibody from the hydrogel persisted for over 12 days in vitro, which can be further accelerated by the addition of proteinase K. A combination antitumor strategy was developed by simultaneous, sustained delivery of Dox and the dual ICB antibodies through peritumoral injection of the multiple-agent co-loaded hydrogel into mice bearing B16F10 melanoma. The treatment with Dox/aCTLA-4/aPD-1 co-loaded hydrogel demonstrated markedly enhanced tumor inhibition efficacy, significantly extended animal survival time and strengthened antitumor immune response, compared to the mixed drug solutions or hydrogel containing Dox or antibodies. Moreover, the localized treatments with the hydrogel-based formulations showed no obvious systemic side effects. Additionally, the treatment with Dox/aCTLA-4/aPD-1 co-loaded hydrogel following tumor resection surgery achieved significantly increased efficiency in inhibiting tumor reoccurrence and extending animal survival time. In summary, the hydrogels encapsulating chemotherapeutics and dual ICB antibodies can serve as a promising candidate as depots for antitumor combination therapy and the prevention of post-operative tumor reoccurrence.

## Data Availability

The data supporting the study is available from the authors on request.

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
