# Peer review of "Injectable Polypeptide Hydrogel Depots Containing Dual Immune Checkpoint Inhibitors and Doxorubicin for Improved Tumor Immunotherapy and Post-Surgical Tumor Treatment"

_pharmaceutics, 2023, doi:10.3390/pharmaceutics15020428_

Round 1

Reviewer 1 Report

Reviewers’ comments for the Manuscript ID:

The manuscript title:Injectable Polypeptide Hydrogel Depots Containing Dual Immune Checkpoint Inhibitors and Chemotherapeutics for Improved Tumor Immunotherapy and Post-Surgical Tumor Treatment”.

In the current manuscript, authors described the combined use of dual immune checkpoint blockade (ICB) antibodies, anti-cytotoxic T-lymphocyte-associated protein 4 (aCTLA-4), anti-programmed cell death protein 1 (aPD-1), and a chemotherapy drug, doxorubicin (Dox), into a thermo-gelling methoxy poly (ethylene glycol)-block-poly(γ-ethyl-L- 16 glutamate) (mPEG-PELG) diblock copolymer hydrogel. It is well organized manuscript, results are presented nicely I don’t have anything to criticize, all the results are anticipated according to theory. The only one question I have is, why authors injected the Dox/aCTLA-4/aPD-1 co-loaded hydrogel at tumor site (peritumorally)?,  did authors also tried other route of injections such as intraperitonially (IP) or  subcutaneous injections?, it  would be better to see toxicity and drug distribution when injected with either of these routes and it will be translatable to humans. Peritumorally injections are not possible in humans right?

Author Response

Reviewer 1:

In the current manuscript, authors described the combined use of dual immune checkpoint blockade (ICB) antibodies, anti-cytotoxic T-lymphocyte-associated protein 4 (aCTLA-4), anti-programmed cell death protein 1 (aPD-1), and a chemotherapy drug, doxorubicin (Dox), into a thermo-gelling methoxy poly (ethylene glycol)-block-poly(γ-ethyl-L- 16 glutamate) (mPEG-PELG) diblock copolymer hydrogel. It is well organized manuscript, results are presented nicely I don’t have anything to criticize, all the results are anticipated according to theory.

1. The only one question I have is, why authors injected the Dox/aCTLA-4/aPD-1 co-loaded hydrogel at tumor site (peritumorally)?, did authors also tried other route of injections such as intraperitonially (IP) or subcutaneous injections?, it would be better to see toxicity and drug distribution when injected with either of these routes and it will be translatable to humans. Peritumorally injections are not possible in humans right?.

Our reply: Thanks a lot for your comments. Compared with that of intraperitoneal injection and subcutaneous injection, the administration of drug-loaded hydrogel by peritumoral injection can release therapeutic agents locally and continuously at the position near the tumor, which may be beneficial for increasing drug concentration at the tumor sites while reducing systemic side-effects (Sci. Transl. Med. 2018, 10, eaan3682; Biomaterials 2018, 159, 119). Therefore, in this study, we administrated the Dox/aCTLA-4/aPD-1 co-loaded hydrogel at tumor site through peritumoral injection, instead of intraperitoneal and subcutaneous injection.

In terms of clinical cancer therapy, the primary strategy is still tumor resection surgery followed by chemotherapy and/or radiotherapy, since residual tumor cells may exist after the surgery. Therefore, to mimic the clinical application in our study, the post-operative tumor-bearing mice model was established. The multiple drugs co-loaded hydrogel was then injected to the surgical site of the mice to prevent tumor reoccurrence. Thus, the drugs can be released at the site near the residual tumor cells, which is also similar to the tumor treatment by peritumoral injection (Biomaterials 2019, 219, 119182).

Reviewer 2 Report

Chen et al. developed injectable hydrogel based on a thermo-gelling mPEG-PELG diblock copolymer for co-delivery of dual immune checkpoint blockade (ICB) antibodies and chemotherapeutic drug doxorubicin (DOX) for synergistic chemo-immunotherapy. The manuscript is generally well written. I only have minor suggestions:

1/ Title: as the chemotherapeutic drug being used in the study was doxorubicin, the word “chemotherapeutics” in the title can be directly replaced with “doxorubicin”.

2/ Line 147: the method to remove solution could be stated.

3/ 2.7 In vitro drug release: there is a lack of information on how to measure the loading efficiency of Dox and IgG. Was any purification applied after loading? or was it assumed that both Dox and IgG were all loaded 100%?

4/ Result and discussion: more discussions to compare the result with current studies from the past 3-5 years can be added.

Author Response

Reviewer 2:

Chen et al. developed injectable hydrogel based on a thermo-gelling mPEG-PELG diblock copolymer for co-delivery of dual immune checkpoint blockade (ICB) antibodies and chemotherapeutic drug doxorubicin (DOX) for synergistic chemo-immunotherapy. The manuscript is generally well written. I only have minor suggestions:

1. Title: as the chemotherapeutic drug being used in the study was doxorubicin, the word “chemotherapeutics” in the title can be directly replaced with “doxorubicin”.

Our reply: Thanks a lot for your suggestion. The word “chemotherapeutics” in the title has been changed to “doxorubicin” with the new title in the revised manuscript as follows:

“Injectable Polypeptide Hydrogel Depots Containing Dual Immune Checkpoint Inhibitors and Doxorubicin for Improved Tumor Immunotherapy and Post-Surgical Tumor Treatment”.

2. Line 147: the method to remove solution could be stated.

Our reply: Thanks a lot for your comments. The solution was removed by gently sucking out from the vial with the pipette and residual solution at the surface of sample was further removed by filter paper. We have supplemented the information in the section 2.6 of the revised manuscript.

3. 2.7 In vitro drug release: there is a lack of information on how to measure the loading efficiency of Dox and IgG. Was any purification applied after loading? or was it assumed that both Dox and IgG were all loaded 100%?

Our reply: Thanks a lot for your comments. To prepare the Dox and IgG co-loaded hydrogel for the in vitro drug release, the precursor polymer solution was mixed with Dox and IgG at 0 °C, and the drug-loaded hydrogel was formed through a sol-gel phase transition in situ by simply increasing the temperature to 37 °C. The drug release test was started by direct addition of release medium to the vial containing drug-loaded hydrogel, and no further purification was applied after loading. Therefore, it was assumed that both Dox and IgG were loaded at 100%.

4. Result and discussion: more discussions to compare the result with current studies from the past 3-5 years can be added.

Our reply: Thanks a lot for your comments. We have added more discussions on comparing the results of current study with the studies of recent years in the revised manuscript. The added discussions are as follows.

In recent studies, hydrogel-based local delivery systems have been investigated for the co-delivery of chemotherapeutics and single ICB antibody, aPD-1 or anti-PD-L1, as a strategy for local chemo-immunotherapy. It has been found that the chemotherapy-mediated ICD of tumor cells could promote the antitumor immune response when combining with ICB blockade. Moreover, some agents for modulating the immunosuppressive tumor micro-environment, e.g. indoleamine-(2,3)-dioxygenase (IDO) inhibitors, were further incorporated into the hydrogel depots for improving the antitumor efficacy. In this study, a new formulation of hydrogel depot co-loaded with dual ICB antibodies, aCTLA-4 and aPD-1, and a chemotherapeutics was developed for topical antitumor chemo-immunotherapy. The sustained release of aCTLA-4 at tumor site can strengthen the antigen presentation of DCs to T cells, and the simultaneous delivery of aPD-1 is able to enhance the subsequent tumor recognition and killing. Based on the treatment of B16F10 melanoma-bearing mice and post-operative mice models, the Dox/aCTLA-4/aPD-1 co-loaded hydrogel showed significantly increased antitumor immune response, enhanced tumor inhibition efficacy and prolonged animal survival time. Additionally, it was demonstrated that the local co-delivery of the ICB antibodies and chemotherapeutics caused no obvious systemic side effects, compared to the risk of severe irAEs by systemic administration of ICB antibodies.

Reviewer 3 Report

1.     The abstract should be refined and shown the most important experiment results or it should be presented in a findings-oriented format in which the most important results and conclusions are summarized.

2.     The authors should highlight the novelty of the manuscript and what is new in their work compared to other published literature.

3.     In results and discussion section, I recommend more profound discussion about the experiment results instead of the simple presentation about the experiment principle and results.

4.     Authors should add Kinetic of release.

5.     Conclusion part is too weak, please improve.

Author Response

Reviewer 3:

Chen et al. developed injectable hydrogel based on a thermo-gelling mPEG-PELG diblock copolymer for co-delivery of dual immune checkpoint blockade (ICB) antibodies and chemotherapeutic drug doxorubicin (DOX) for synergistic chemo-immunotherapy. The manuscript is generally well written. I only have minor suggestions:

1. The abstract should be refined and shown the most important experiment results or it should be presented in a findings-oriented format in which the most important results and conclusions are summarized.

Our reply: Thanks a lot for your comments. We have refined the abstract in the revised manuscript, and the most important experiment results are shown in the abstract.

2. The authors should highlight the novelty of the manuscript and what is new in their work compared to other published literature.

Our reply: Thanks a lot for your comments. We have highlighted the novelty of our work compared to other published literatures. The revision in the revised manuscript is as follows.

In recent studies, hydrogel-based local delivery systems have been investigated for the co-delivery of chemotherapeutics and single ICB antibody, aPD-1 or anti-PD-L1, as a strategy for local chemo-immunotherapy. It has been found that the chemotherapy-mediated ICD of tumor cells could promote the antitumor immune response when combining with ICB blockade. Moreover, some agents for modulating the immunosuppressive tumor micro-environment, e.g. indoleamine-(2,3)-dioxygenase (IDO) inhibitors, were further incorporated into the hydrogel depots for improving the antitumor efficacy. In this study, a new formulation of hydrogel depot co-loaded with dual ICB antibodies, aCTLA-4 and aPD-1, and a chemotherapeutics was developed for topical antitumor chemo-immunotherapy. The sustained release of aCTLA-4 at tumor site can strengthen the antigen presentation of DCs to T cells, and the simultaneous delivery of aPD-1 is able to enhance the subsequent tumor recognition and killing. Based on the treatment of B16F10 melanoma-bearing mice and post-operative mice models, the Dox/aCTLA-4/aPD-1 co-loaded hydrogel showed significantly increased antitumor immune response, enhanced tumor inhibition efficacy and prolonged animal survival time. Additionally, it was demonstrated that the local co-delivery of the ICB antibodies and chemotherapeutics caused no obvious systemic side effects, compared to the risk of severe irAEs by systemic administration of ICB antibodies.

3. In results and discussion section, I recommend more profound discussion about the experiment results instead of the simple presentation about the experiment principle and results.

Our reply: Thanks a lot for your comments. We have supplied more discussions on the experiment results in the revised manuscript.

The supplied discussions are listed as follows.

The partial dehydration of mPEG segments may result in enhanced chain entanglement and aggregation of the mPEG-PELG micelles in aqueous solution, and the β-sheet con-formation of the polypeptide blocks facilitates the formation of intermolecular hydrogen-bonding. These effects promote the formation of physical crosslinking network and hydrogel.

It is known that there are different enzymes, such as cathepsin B, cathepsin C and elastase, in the subcutaneous layer of mammals. Therefore, the degradation of the hydrogel in the subcutaneous layer of rats may be accelerated by the enzymatic hydrolysis in vivo.

Compared to systemic administration, the local and sustained release of Dox and ICB antibodies at tumor sites can markedly reduce the blood drug concentration and drug distribution in normal tissues. Thus, low systemic toxicity was observed for the local treatments.”

IL-2 is a cytokine that undertake immunoregulatory functions on T cells, NK cells and NK-T cells. IL-2 plays a role in promoting the proliferation of lymphocytes including T cells and NK cells. TNF-α is able to exert antitumor effects by direct showing cytotoxicity on tumor cells and inducing antitumor immune response. IFN-γ is a crucial cytokine for cell-mediated immunity, which plays key roles in stimulating antigen presentation, cytokine secretion, as well as activation of macrophages, NK cells and neutrophils.

4. Authors should add Kinetic of release.

Our reply: Thanks a lot for your comments. We fitted the in vitro drug release data in the first 6 days to different kinetic models, including zero-order model, first-order model, Korsmeyer-Peppas model and Higuchi model (Figure S5) (Materials 2022, 15, 8711; Acta Poloniae Pharmaceutica - Drug Research 2010, 67, 217-223; International Journal of Pharmaceutics 2011, 418, 6-12). Through comparing the R-squared value, the Dox release profile from the hydrogel in PBS showed a relatively better degree of fitting to Korsmeyer-Peppas model. It is noteworthy that the R-squared value was reduced for Dox release in the presence of proteinase K. Additionally, the IgG release from the hydrogel did not fit well with these kinetic models, which may be attributed to the fact that the IgG release behavior was influenced by complicated interactions between the protein and hydrogel network. (Journal of Drug Delivery Science and Technology 2021, 63, 102483)

We have added the results in the revised manuscript.

5. Conclusion part is too weak, please improve.

Our reply: Thanks a lot for your comments. We have improved the conclusion part in the revised manuscript.
